# Learning to Model the Tail

**Yu-Xiong Wang**       **Deva Ramanan**       **Martial Hebert**
Robotics Institute, Carnegie Mellon University
{yuxiongw,dramanan,hebert}@cs.cmu.edu

## Abstract

We describe an approach to learning from long-tailed, imbalanced datasets that
are prevalent in real-world settings. Here, the challenge is to learn accurate "few-
shot" models for classes in the tail of the class distribution, for which little data
is available. We cast this problem as transfer learning, where knowledge from
the data-rich classes in the head of the distribution is transferred to the data-poor
classes in the tail. Our key insights are as follows. First, we propose to transfer
*meta*-knowledge about *learning-to-learn* from the head classes. This knowledge is
encoded with a meta-network that operates on the space of model parameters, that
is trained to predict many-shot model parameters from few-shot model parameters.
Second, we transfer this meta-knowledge in a *progressive* manner, from classes
in the head to the "body", and from the "body" to the tail. That is, we transfer
knowledge in a gradual fashion, regularizing meta-networks for few-shot regression
with those trained with more training data. This allows our final network to capture
a notion of *model dynamics*, that predicts how model parameters are likely to
change as more training data is gradually added. We demonstrate results on
image classification datasets (SUN, Places, and ImageNet) tuned for the long-tailed
setting, that significantly outperform common heuristics, such as data resampling
or reweighting.

## 1   Motivation

Deep convolutional neural networks (CNNs) have revolutionized the landscape of visual recognition,
through the ability to learn "big models" with hundreds of millions of parameters [1, 2, 3, 4]. Such
models are typically learned with *artificially balanced* datasets [5, 6, 7], in which objects of different
classes have approximately evenly distributed, very large number of human-annotated images. In
real-world applications, however, visual phenomena follow a long-tailed distribution as shown in
Fig. 1, in which the number of training examples per class varies significantly from hundreds or
thousands for head classes to as few as one for tail classes [8, 9, 10].

**Long-tail:** Minimizing the skewed distribution by collecting more tail examples is a notoriously
difficult task when constructing datasets [11, 6, 12, 10]. Even those datasets that are balanced along
one dimension still tend to be imbalanced in others [13]; *e.g.*, balanced scene datasets still contain
long-tail sets of objects [14] or scene subclasses [8]. This *intrinsic* long-tail property poses a multitude
of open challenges for recognition in the wild [15], since the models will be largely dominated by
those few head classes while degraded for many other tail classes. Rebalancing training data [16, 17]
is the most widespread state-of-the-art solution, but this is *heuristic and suboptimal* — it merely
generates redundant data through over-sampling or loses critical information through under-sampling.

**Head-to-tail knowledge transfer:** An attractive alternative is to *transfer* knowledge from data-rich
head classes to data-poor tail classes. While transfer learning from a source to target task is a well
studied problem [18, 19], by far the most common approach is fine-tuning a model pre-trained on the
source task [20]. In the long-tailed setting, this fails to provide any noticeable improvement since

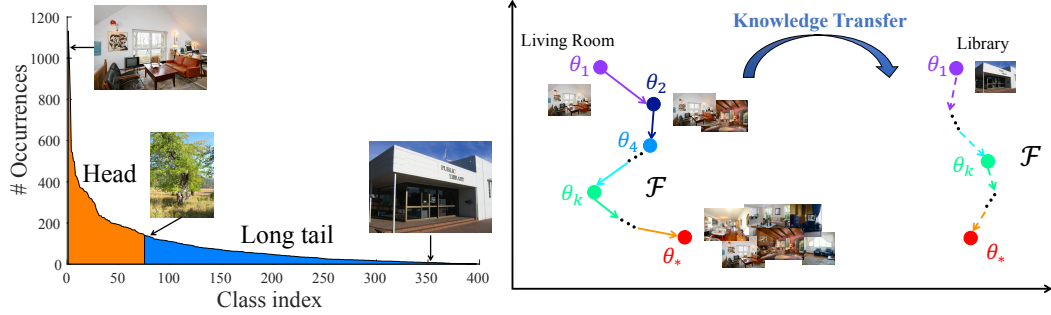

(a) Long-tail distribution on the SUN-397 dataset.　　　(b) Knowledge transfer from head to tail classes.

Figure 1: Head-to-tail knowledge transfer in model space for long-tail recognition. Fig. 1a shows the number of examples by scene class on SUN-397 [14], a representative dataset that follows an intrinsic long-tailed distribution. In Fig. 1b, from the data-rich head classes (*e.g.*, living rooms), we introduce a meta-learner $\mathcal{F}$ to learn the model dynamics — a series of transformations (denoted as solid lines) that represents how few $k$-shot models $\theta_k$ start from $\theta_1$ and gradually evolve to the underlying many-shot models $\theta_*$ trained from large sets of samples. The model parameters $\theta$ are visualized as points in the "dual" model (parameter) space. We leverage the model dynamics as prior knowledge to facilitate recognizing tail classes (*e.g.*, libraries) by hallucinating their model evolution trajectories (denoted as dashed lines).

pre-training on the head is quite similar to training on the unbalanced long-tailed dataset (which is dominated by the head) [10].

**Transferring meta-knowledge:** Inspired by the recent work on meta-learning [21, 22, 23, 24, 25, 26], we instead transfer meta-level knowledge about *learning to learn* from the head classes. Specifically, we make use of the approach of [21], which describes a method for learning from small datasets (the "few-shot" learning problem) through estimating a generic model transformation. To do so, [21] learns a meta-level network that operates on the space of model parameters, which is specifically trained to *regress* many-shot model parameters (trained on large datasets) from few-shot model parameters (trained on small datasets). Our meta-level regressor, which we call *MetaModelNet*, is trained on classes from the head of the distribution and then applied to those from the tail. As an illustrative example in Fig. 1, consider learning scene classifiers on a long-tailed dataset with many living-rooms but few outside libraries. We learn both many-shot and few-shot living-room models (by subsampling the training data as needed), and train a regressor that maps between the two. We can then apply the regressor on few-shot models of libraries learned from the tail.

**Progressive transfer:** The above description suggests that we need to split up a long-tailed training set into a distinct set of source classes (the head) and target classes (the tail). This is most naturally done by thresholding the number of training examples per class. But what is the correct threshold? A high threshold might result in a meta-network that simply acts as an identity function, returning the input set of model parameters. This certainly would not be useful to apply on few-shot models. Similarly, a low threshold may not be useful when regressing from many-shot models. Instead, we propose a "continuous" strategy that builds multiple regressors across a (logarithmic) *range* of thresholds (*e.g.*, 1-shot, 2-shot, 4-shot regressors, *etc.*), corresponding to different head-tail splits. Importantly, these regressors can be efficiently implemented with a *single, chained* MetaModelNet that is naturally regularized with residual connections, such that the 2-shot regressor need only predict model parameters that are fed into the 4-shot regressor, and so on (until the many-shot regressor that defaults to the identity). By doing so, MetaModelNet encodes a trajectory over the space of model parameters that captures their evolution with increasing sample sizes, as shown in Fig. 1b. Interestingly, such a network is naturally trained in a *progressive* manner from the head towards the tail, effectively capturing the gradual dynamics of transferring meta-knowledge from data-rich to data-poor regimes.

**Model dynamics:** It is natural to ask what kind of dynamics are learned by MetaModelNet — how can one consistently predict how model parameters will change with more training data? We posit that the network learns to capture implicit *data augmentation* — for example, given a 1-shot model trained with a single image, the network may learn to implicitly add rotations of that single image. But rather

than explicitly creating data, MetaModelNet predicts their impact on the learned model parameters. Interestingly, past work tends to apply the same augmentation strategies across all input classes. But perhaps different classes should be augmented in different ways — *e.g.*, churches maybe viewed from consistent viewpoints and should not be augmented with out-of-plane rotations. MetaModelNet learns *class-specific* transformations that are smooth across the space of models — *e.g.*, classes with similar model parameters tend to transform in similar ways (see Fig. 1b and Fig. 4 for more details).

**Our contributions** are three-fold. (1) We analyze the *dynamics* of how model parameters evolve when given access to more training examples. (2) We show that a single meta-network, based on deep residual learning, can learn to accurately predict such dynamics. (3) We train such a meta-network on long-tailed datasets through a recursive approach that gradually transfers meta-knowledge learned from the head to the tail, significantly improving long-tail recognition on a broad range of tasks.

## 2 Related Work

A widespread yet suboptimal strategy is to resample and rebalance training data in the presence of the long tail, either by sampling examples from the rare classes more frequently [16, 17], or reducing the number of examples from the common classes [27]. The former generates redundancy and quickly runs into the problem of over-fitting to the rare classes, whereas the latter loses critical information contained within the large-sample sets. An alternative practice is to introduce additional weights for different classes, which, however, makes optimization of the models very difficult in the large-scale recognition scenarios [28].

Our underlying assumption that model parameters across different classes share similar dynamics is somewhat common in meta-learning [21, 22, 25]. While [22, 25] consider the dynamics during stochastic gradient descent (SGD) optimization, we address the dynamics as more training data is gradually made available. In particular, the model regression network from [21] empirically shows a generic nonlinear transformation from small-sample to large-sample models for different types of feature spaces and classifier models. We extend [21] for long-tail recognition by introducing a single network that can model transformations across different samples sizes. To train such a network, we introduce recursive algorithms for head-to-tail transfer learning and architectural modifications based on deep residual networks (that ensure that transformations of large-sample models default to the identity).

Our approach is broadly related to different meta-learning concepts such as learning-to-learn, transfer learning, and multi-task learning [29, 30, 18, 31]. Such approaches tend to learn shared structures from a set of relevant tasks and generalize to novel tasks. Specifically, our approach is inspired by early work on parameter prediction that modifies the weights of one network using another [32, 33, 34, 35, 36, 37, 38, 26, 39]. Such techniques have also been recently explored in the context of regressing classifier weights from training sample [40, 41, 42]. From an optimization perspective, our approach is related to work on learning to optimize, which replaces hand-designed update rules (*e.g.,* SGD) with a learned update rule [22, 24, 25].

The most related formulation is that of one/few-shot learning [43, 44, 45, 46, 47, 48, 21, 49, 35, 50, 23, 51, 52, 53, 54, 55, 56]. Past work has explored strategies of using the common knowledge captured among a set of one-shot learning tasks during meta-training for a novel one-shot learning problem [52, 25, 35, 53]. These techniques, however, are typically developed for a fixed set of few-shot tasks, in which each class has the same, fixed number of training samples. They appear difficult to generalize to novel tasks with a wide range of sample sizes, the hallmark of long-tail recognition.

## 3 Head-to-Tail Meta-Knowledge Transfer

Given a long-tail recognition task of interest and a base recognition model such as a deep CNN, our goal is to transfer knowledge from the data-rich head to the data-poor tail classes. As shown in Fig. 1, knowledge is represented as trajectories in model space that capture the evolution of parameters with more and more training examples. We train a meta-learner (MetaModelNet) to learn such model dynamics from head classes, and then "hallucinate" the evolution of parameters for the tail classes. To simplify exposition, we first describe the approach for a fixed split of our training dataset into a head and tail. We then generalize the approach to multiple splits.

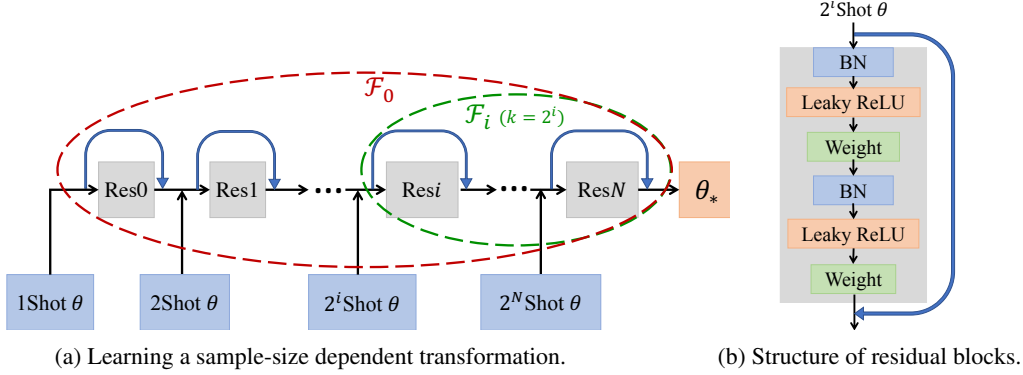

(a) Learning a sample-size dependent transformation.

(b) Structure of residual blocks.

Figure 2: MetaModelNet architecture for learning model dynamics. We instantiate MetaModelNet as a deep residual network with residual blocks $i = 0, 1, \ldots, N$ in Fig. 2a, which accepts few-shot model parameters $\theta$ (trained on small datasets across a logarithmic range of sample sizes $k$, $k = 2^i$) as (multiple) inputs and regresses them to many-shot model parameters $\theta_*$ (trained on large datasets) as output. The skip connections ensure the identity regularization. $\mathcal{F}_i$ denotes the meta-learner that transforms (regresses) $k$-shot $\theta$ to $\theta_*$. Fig. 2b shows the structure of the residual blocks. Note that the meta-learners $\mathcal{F}_i$ for different $k$ are derived from this single, chained meta-network, with nested circles (subnetworks) corresponding to $\mathcal{F}_i$.

## 3.1 Fixed-size model transformations

Let us write $H_t$ for the "head" training set of $(x, y)$ data-label pairs constructed by assembling those classes for which there exist more than $t$ training examples. We will use $H_t$ to learn a meta-network thats maps few-shot model parameters to many-shot parameters, and then apply this network on few-shot models from the tail classes. To do so, we closely follow the model regression framework from [21], but introduce notation that will be useful later. Let us write a base learner as $g(x; \theta)$ as a feedforward function $g(\cdot)$ that processes an input sample $x$ given parameters $\theta$. We first learn a set of "optimal" model parameters $\theta_*$ by tuning $g$ on $H_t$ with a standard loss function. We also learn few-shot models by randomly sampling a smaller fixed number of examples per class from $H_t$. We then train a meta-network $\mathcal{F}(\cdot)$ to map or regress the few-shot parameters to $\theta_*$.

**Parameters:** In principle, $\mathcal{F}(\cdot)$ applies to model parameters from multiple CNN layers. Directly regressing parameters from all layers is, however, difficult to do because of the larger number of parameters. For example, recent similar methods for meta-learning tend to restrict themselves to smaller toy networks [22, 25]. For now, we focus on parameters from the last fully-connected layer for a single class — e.g., $\theta \in \mathbb{R}^{4096}$ for an AlexNet architecture. This allows us to learn regressors that are shared across classes (as in [21]), and so can be applied to any individual test class. This is particularly helpful in the long-tailed setting, where the number of classes in the tail tends to outnumber the head. Later we will show that (nonlinear) fine-tuning of the "entire network" during head-to-tail transfer can further improve performance.

**Loss function:** The meta-network $\mathcal{F}(\cdot)$ is itself parameterized with weights $w$. The objective function for each class is:

$$\sum_{\theta \in k\text{Shot}(H_t)} \left\{ ||\mathcal{F}(\theta; w) - \theta_*||^2 + \lambda \sum_{(x,y) \in H_t} \text{loss}\Big(g\big(x; \mathcal{F}(\theta; w)\big), y\Big) \right\}. \tag{1}$$

The final loss is averaged over all the head classes and minimized with respect to $w$. Here, $k\text{Shot}(H_t)$ is the set of few-shot models learned by subsampling $k$ examples per class from $H_t$, and loss refers to the performance loss used to train the base network (e.g., cross-entropy). $\lambda > 0$ is the regularization parameter used to control the trade-off between the two terms. [21] found that the performance loss was useful to learn regressors that maintained high accuracy on the base task. This formulation can be viewed as an extension to those in [21, 25]. With only the performance loss, Eqn. (1) reduces to the loss function in [25]. When the performance loss is evaluated on the subsampled set, Eqn. (1) reduces to the loss function in [21].

**Training:** What should be the value of $k$, for the $k$-shot models being trained? One might be tempted to set $k = t$, but this implies that there will be some head classes near the cutoff that have only $t$ training examples, implying $\theta$ and $\theta_*$ will be identical. To ensure that a meaningful mapping is learned, we set

$$k = t/2.$$

In other terms, we intentionally learn *very*-few-shot models to ensure that target model parameters are sufficiently more general.

## 3.2   Recursive residual transformations

We wish to apply the above module on all possible head-tail splits of a long-tailed training set. To do so, we extend the above approach in three crucial ways:

- (Sample-size dependency) Generate a sequence of different meta-learners $\mathcal{F}_i$ each tuned for a specific $k$, where $k = k(i)$ is an increasing function of $i$ (that will be specified shortly). Through a straightforward extension, prior work on model regression [21] learns a single fixed meta-learner for all the $k$-shot regression tasks.

- (Identity regularization) Ensure that the meta-learner defaults to the identity function for large $i$: $\mathcal{F}_i \to \mathcal{I}$ as $i \to \infty$.

- (Compositionality) Compose meta-learners out of each other: $\forall i < j, \mathcal{F}_i(\theta) = \mathcal{F}_j\Big(\mathcal{F}_{ij}(\theta)\Big)$

  where $\mathcal{F}_{ij}$ is the regressor that maps between $k(i)$-shot and $k(j)$-shot models.

Here we dropped the explicit dependence of $\mathcal{F}(\cdot)$ on $w$ for notational simplicity. These observations emphasize the importance of (1) the identity regularization and (2) sample-size dependent regressors for long-tailed model transfer. We operationalize these extensions with a recursive residual network:

$$\mathcal{F}_i(\theta) = \mathcal{F}_{i+1}\Big(\theta + f(\theta; w_i)\Big), \tag{2}$$

where $f$ denotes a residual block parameterized by $w_i$ and visualized in Fig. 2b. Inspired by [57, 21], $f$ consists of batch normalization (BN) and leaky ReLU as pre-activation, followed by fully-connected weights. By construction, each residual block transforms an input $k(i)$-shot model to a $k(i+1)$-shot model. The final MetaModelNet can be efficiently implemented through a chained network of $N+1$ residual blocks, as shown in Fig. 2a. By feeding in a few-shot model at a particular block, we can derive any meta-learner $\mathcal{F}_i$ from the central underlying chain.

## 3.3   Training

Given the network structure defined above, we now describe an efficient method for training based on two insights. (1) The recursive definition of MetaModelNet suggests a recursive strategy for training. We begin with the *last* block and train it with the *largest* threshold (*e.g.*, those few classes in the head with many examples). The associated $k$-shot regressor should be easy to learn because it is similar to an identity mapping. Given the learned parameters for the last block, we then train the next-to-last block, and so on. (2) Inspired by the general observation that recognition performance improves *on a logarithmic scale* as the number of training samples increases [8, 9, 58], we discretize blocks accordingly, to be tuned for 1-shot, 2-shot, 4-shot, ... recognition. In terms of notation, we write the recursive training procedure as follows. We iterate over blocks $i$ from $N$ to 0, and for each $i$:

- Using Eqn. (1), train parameters of the residual block $w_i$ on the head split $H_t$ with $k$-shot model regression, where $k = 2^i$ and $t = 2k = 2^{i+1}$.

The above "back-to-front" training procedure works because whenever block $i$ is trained, all subsequent blocks $(i+1, \ldots, N)$ have already been trained. In practice, rather than holding all subsequent blocks fixed, it is natural to fine-tune them while training block $i$. One approach might be fine-tuning them on the current $k = 2^i$-shot regression task being considered at iteration $i$. But because Meta-ModelNet will be applied across a wide range of $k$, we fine-tune blocks in a *multi-task* manner across the current viable range of $k = (2^i, 2^{i+1}, \ldots, 2^N)$ at each iteration $i$.

### 3.4 Implementation details

We learn the CNN models on the long-tailed recognition datasets in different scenarios: (1) using a CNN pre-trained on ILSVRC 2012 [1, 59, 60] as the off-the-shelf feature; (2) fine-tuning the pre-trained CNN; and (3) training a CNN from scratch. We use ResNet152 [4] for its state-of-the-art performance and use ResNet50 [4] and AlexNet [1] for their easy computation.

When training the residual block $i$, we use the corresponding threshold $t$ and obtain $C_t$ head classes. We generate the $C_t$-way many-shot classifiers on $H_t$. For few-shot models, we learn $C_t$-way $k$-shot classifiers on random subsets of $H_t$. Through random sampling, we generate $S$ model mini-batches and each model mini-batch consists of $C_t$ weight vector pairs. In addition, to minimize the loss function (1), we randomly sample 256 image-label pairs as a data mini-batch from $H_t$.

We then use Caffe [59] to train our MetaModelNet on the generated model and data mini-batches based on standard SGD. $\lambda$ is cross-validated. We use $0.01$ as the negative slope for leaky ReLU. Computation is naturally divided into two stages: (1) training a collection of few/many-shot models and (2) learning MetaModelNet from those models. (2) is equivalent to progressively learning a nonlinear regressor. (1) can be made efficient because it is naturally parallelizable across models, and moreover, many models make use of only small training sets.

## 4 Experimental Evaluation

In this section, we explore the use of our MetaModelNet on long-tail recognition tasks. We begin with extensive evaluation of our approach on scene classification of the SUN-397 dataset [14], and address the meta-network variations and different design choices. We then visualize and empirically analyze the learned model dynamics. Finally, we evaluate on the challenging large-scale, scene-centric Places [7] and object-centric ImageNet datasets [5] and show the generality of our approach.

### 4.1 Evaluation and analysis on SUN-397

**Dataset and task:** We start our evaluation by fine-tuning a pre-trained CNN on SUN-397, a medium-scale, long-tailed dataset with 397 classes and 100–2,361 images per class [14]. To better analyze trends due to skewed distributions, we carve out a more extreme version of the dataset. Following the experimental setup in [61, 62, 63], we first randomly split the dataset into train, validation, and test parts using $50\%$, $10\%$, and $40\%$ of the data, respectively. The distribution of classes is uniform across all the three parts. We then randomly discard 49 images per class for the train part, leading to a long-tailed training set with 1–1,132 images per class (median 47). Similarly, we generate a small long-tailed validation set with 1–227 images per class (median 10), which we use for learning hyper-parameters. We also randomly sample 40 images per class for the test part, leading to a balanced test set. We report 397-way multi-class classification accuracy averaged over all classes.

#### 4.1.1 Comparison with state-of-the-art approaches

We first focus on fine-tuning the classifier module while freezing the representation module of a pre-trained ResNet152 CNN model [4, 63] for its state-of-the-art performance. Using MetaModelNet, we learn the model dynamics of the classifier module, *i.e.,* how the classifier weight vectors change during fine-tuning. Following the design choices in Section 3.2, our MetaModelNet consists of 7 residual blocks. For few-shot models, we generate $S = 1000$ 1-shot, $S = 500$ 2-shot, and $S = 200$ 4-shot till 64-shot models from the head classes for learning MetaModelNet. At test time, given the weight vectors of all the classes learned through fine-tuning, we feed them as inputs to the different residual blocks according to their training sample size of the corresponding class. We then "hallucinate" the dynamics of these weight vectors and use the outputs of MetaModelNet to modify the parameters of the final recognition model as in [21].

**Baselines:** In addition to the "plain" baseline that fine-tunes on the target data following the standard practice, we compare against three state-of-the-art baselines that are widely used to address the imbalanced distributions. (1) Over-sampling [16, 17], which uses the balanced sampling via label shuffling as in [16, 17]. (2) Under-sampling [27], which reduces the number of samples per class to 47 at most (the median value). (3) Cost-sensitive [28], which introduces additional weights in the loss function for each class with inverse class frequency. For a fair comparison, fine-tuning is performed

| Method | Plain [4] | Over-Sampling [16, 17] | Under-Sampling [27] | Cost-Sensitive [28] | MetaModelNet (Ours) |
|---|---|---|---|---|---|
| Acc (%) | 48.03 | 52.61 | 51.72 | 52.37 | **57.34** |

Table 1: Performance comparison between our MetaModelNet and state-of-the-art approaches for long-tailed scene classification when fine-tuning the pre-trained ILSVRC ResNet152 on the SUN-397 dataset. We focus on learning the model dynamics of the classifier module while freezing the CNN representation module. By benefiting from the learned generic model dynamics from head classes, ours significantly outperforms all the baselines for the long-tail recognition.

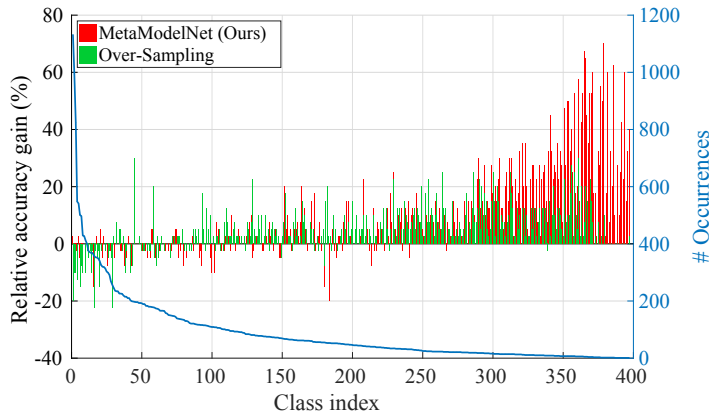

Figure 3: Detailed per class performance comparison between our MetaModelNet and the state-of-the-art over-sampling approach for long-tailed scene classification on the SUN-397 dataset. X-axis: class index. Y-axis (Left): per class classification accuracy improvement relative to the plain baseline. Y-axis (Right): number of training examples. Ours significantly improves for the few-shot tail classes.

for around 60 epochs using SGD with an initial learning rate of $0.01$, which is reduced by a factor of 10 around every 30 epochs. All the other hyper-parameters are the same for all approaches.

Table 1 summarizes the performance comparison averaged over all classes and Fig. 3 details the per class comparison. Table 1 shows that our MetaModelNet provides a promising way of encoding the shared structure across classes *in model space*. It outperforms existing approaches for long-tail recognition by a large margin. Fig. 3 shows that our approach significantly improves accuracy in the tail.

### 4.1.2 Ablation analysis

We now evaluate variations of our approach and provide ablation analysis. Similar as in Section 4.1.1, we use ResNet152 in the first two sets of experiments and only fine-tune the classifier module. In the last set of experiments, we use ResNet50 [4] for easy computation and fine-tune through the entire network. Tables 2 and 3 summarize the results.

**Sample-size dependent transformation and identity regularization:** We compare to [21], which learns a single transformation for a variety of sample sizes and $k$-shot models, and importantly, learns a network without identity regularization. For a fair comparison, we consider a variant of MetaModelNet trained on a fixed head and tail split, selected by cross-validation. Table 2 shows that training for a fixed sample size and identity regularization provide a noticeable performance boost (2%).

**Recursive class splitting:** Adding multiple head-tail splits through recursion further improves accuracy by a small but noticeable amount (0.5% as shown in Table 2). We posit that progressive knowledge transfer outperforms the traditional approach because ordering classes by frequency is a natural form of curriculum learning.

**Joint feature fine-tuning and model dynamics learning:** We also explore (nonlinear) fine-tuning of the "entire network" during head-to-tail transfer by jointly learning the classifier dynamics and the feature representation using ResNet50. We explore two approaches as follows. (1) We first fine-tune

| Method | Model Regression [21] | MetaModelNet+Fix Split (Ours) | MetaModelNet+ Recur Split (Ours) |
|---|---|---|---|
| Acc (%) | 54.68 | 56.86 | **57.34** |

Table 2: Ablation analysis of variations of our MetaModelNet. In a fixed head-tail split, ours outperforms [21], showing the merit of learning a sample-size dependent transformation. By recursively partitioning the entire classes into different head-tail splits, our performance is further improved.

| Scenario | Pre-Trained Features | | Fine-Tuned Features (FT) | | |
|---|---|---|---|---|---|
| Method | Plain [4] | MetaModelNet (Ours) | Plain [4] | Fix FT + MetaModelNet (Ours) | Recur FT + MetaModelNet (Ours) |
| Acc (%) | 46.90 | 54.99 | 49.40 | 58.53 | **58.74** |

Table 3: Ablation analysis of joint feature fine-tuning and model dynamics learning on a ResNet50 base network. Though results with pre-trained features underperform those with a deeper base network (ResNet152, the default in our experiments), fine-tuning such features significantly improves results, even outperforming the deeper base network. By progressively fine-tuning the representation during the recursive training of MetaModelNet, performance significantly improves from 54.99% (changing only the classifier weights) to 58.74% (changing the entire CNN).

the whole CNN on the entire long-tailed training dataset, and then learn the classifier dynamics using the fixed, fine-tuned representation. (2) During the recursive head-tail splitting, we fine-tune the entire CNN on the current head classes in $H_t$ (while learning the many-shot parameters $\theta_*$), and then learn classifier dynamics using the fine-tuned features. Table 3 shows that progressively learning classifier dynamics while fine-tuning features performs the best.

## 4.2 Understanding model dynamics

Because model dynamics are highly nonlinear, a theoretical proof is rather challenging and outside the scope of this work. Here we provide some empirical analysis of model dynamics. When analyzing the "dual model (parameter) space", in which models parameters $\theta$ can be viewed as points, Fig. 4 shows that our MetaModelNet learns an approximately-smooth, nonlinear warping of this space that transforms (few-shot) input points to (many-shot) output points. For example, iceberg and mountain scene classes are more similar to each other than to bedrooms. This implies that few-shot iceberg and mountain scene models lie near each other in parameter space, and moreover, they transform in similar ways (when compared to bedrooms). This single meta-network hence encodes class-specific model transformations. We posit that the transformation may capture some form of (class-specific) data-augmentation. Finally, we find that some properties of the learned transformations are quite class-agnostic and apply in generality. Many-shot model parameters tend to have larger magnitudes and norms than few-shot ones (*e.g.*, on SUN-397, the average norm of 1-shot models is 0.53; after transformations through MetaModelNet, the average norm of the output models becomes 1.36). This is consistent with the common empirical observation that classifier weights tend to grow with the amount of training data, showing that they become more confident about their prediction.

## 4.3 Generalization to other tasks and datasets

We now focus on the more challenging, large-scale scene-centric Places [7] and object-centric ImageNet [5] datasets. While we mainly addressed the model dynamics when fine-tuning a pre-trained CNN in the previous experiments, here we train AlexNet models [1] from scratch on the target tasks. Table 4 shows the generality of our approach and shows that MetaModelNets facilitate the recognition of other long-tailed datasets with significantly different visual concepts and distributions.

**Scene classification on the Places dataset:** Places-205 [7] is a large-scale dataset which contains 2,448,873 training images approximately evenly distributed across 205 classes. To generate its long-tailed version and better analyze trends due to skewed distributions, we distribute it according to the distribution of SUN and carve out a more extreme version ($p^2$, or $2\times$ the slope in log-log plot) out of the Places training portion, leading to a long-tailed training set with 5–9,900 images per class (median 73). We use the provided validation portion as our test set with 100 images per class.

**Object classification on the ImageNet dataset:** The ILSVRC 2012 classification dataset [5] contains 1,000 classes with 1.2 million training images (approximately balanced between the classes) and 50K validation images. There are 200 classes used for object detection which are defined as

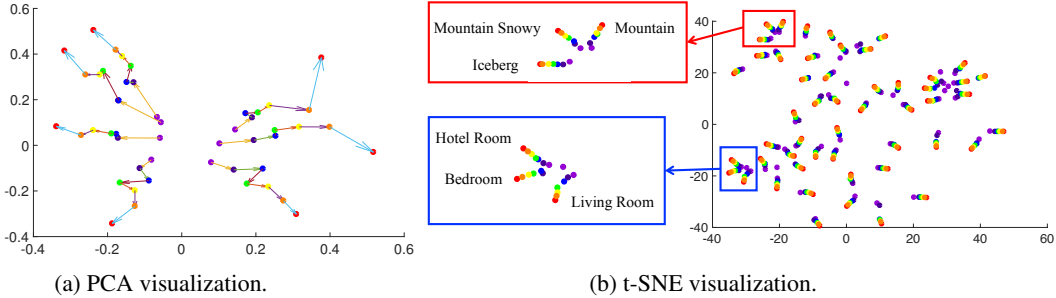

|                       | (a) PCA visualization. | (b) t-SNE visualization. |

Figure 4: Visualizing model dynamics. Recall that $\theta$ is a fixed-dimensional vector of model parameters — *e.g.*, $\theta \in \mathbb{R}^{2048}$ when considering parameters from the last layer of ResNet. We visualize models as points in this "dual" space. Specifically, we examine the evolution of parameters predicted by MetaModelNet with dimensionality reduction — PCA (Fig. 4a) or t-SNE [64] (Fig. 4b). 1-shot models (purple) to many-shot models (red) are plotted in a rainbow order. These visualizations show that MetaModelNet learns an approximately-smooth, nonlinear warping of this space that transforms (few-shot) input points to (many-shot) output points. PCA suggests that many-shot models tend to have larger norms, while t-SNE (which nonlinearly maps nearby points to stay close) suggests that similar semantic classes tend to be close and transform in similar ways, *e.g.,* the blue rectangle encompasses "room" classes while the red rectangle encompasses "wintry outdoor" classes.

| Dataset | Places-205 [7] | | ILSVRC-2012 [5] | |
|---|---|---|---|---|
| Method | Plain [1] | MetaModelNet (Ours) | Plain [1] | MetaModelNet (Ours) |
| Acc (%) | 23.53 | **30.71** | 68.85 | **73.46** |

Table 4: Performance comparisons on long-tailed, large-scale scene-centric Places [7] and object-centric ImageNet [5] datasets. Our MetaModelNets facilitate the long-tail recognition with significantly diverse visual concepts and distributions.

higher-level classes of the original $1,000$ classes. Taking the ILSVRC 2012 classification dataset and merging the $1,000$ classes into the $200$ higher-level classes, we obtain a natural long-tailed distribution.

## 5 Conclusions

In this work we proposed a conceptually simple but powerful approach to address the problem of long-tail recognition through knowledge transfer from the head to the tail of the class distribution. Our key insight is to represent the model dynamics through meta-learning, *i.e.,* how a recognition model transforms and evolves during the learning process when gradually encountering more training examples. To do so, we introduce a meta-network that learns to progressively transfer meta-knowledge from the head to the tail classes. We present several state-of-the-art results on benchmark datasets (SUN, Places, and Imagenet) tuned for the long-tailed setting, that significantly outperform common heuristics, such as data resampling or reweighting.

**Acknowledgments**. We thank Liangyan Gui, Olga Russakovsky, Yao-Hung Hubert Tsai, and Ruslan Salakhutdinov for valuable and insightful discussions. This work was supported in part by ONR MURI N000141612007 and U.S. Army Research Laboratory (ARL) under the Collaborative Technology Alliance Program, Cooperative Agreement W911NF-10-2-0016. DR was supported in part by the National Science Foundation (NSF) under grant number IIS-1618903, Google, and Facebook. We also thank NVIDIA for donating GPUs and AWS Cloud Credits for Research program.

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
