[Reviews · NeurIPS 2017]

Reviewer 1



Summary ------- The paper proposes an approach for transfer learning for multi-class classification problems that aids the learning of categories with few training examples (the categories in the tail of the distribution of numbers of examples per category). It is based on ideas of meta-learning: it builds a (meta-)model of the dynamics that accompany the change in model parameters as more training data is made available to a classifier. Specifically, the proposed approach takes inspiration from existing work on meta-learning [20] but extends it by applying it to CNNs, utilizing deep residual networks as the meta-model, and applying the framework to a general 'long-tail problem' setting in which the number of training examples available is different and not fixed between categories. Experiments are conducted on curated versions of existing datasets (curated such that they exhibit strong long-tail distributions): SUN-397 [13], Places [7], and ImageNet [5]. The performance of the proposed method is demonstrated to be considerably higher than several more adhoc baselines from the literature. Novelty and significance ------------------------ While the proposed method draws inspiration from a variety of existing methods in transfer learning (notably [20]), the particular formulation using deep residual networks as a meta-model, modeling the dynamics of adding progressively more training examples, and handling the case of different numbers of available training examples per category in a unified and automatic way are novel aspects of the paper to my knowledge. These aspects could potentially be useful to a wider range of applications than (image) classification and facilitate the treatment of long-tail problems, as they arise in many real-world application scenarios. Technical correctness --------------------- The proposed method of (meta-)modeling the gradual addition of training examples seems plausible and technically correct. It would, however, be interesting to learn more about the complexity of the proposed approach in terms of training time, since a significant number of individual models has to be trained prior to training the meta-model. Experimental evalution ---------------------- The experimental evaluation seems convincing: it is done on three different existing datasets, compares the proposed approach to existing, more ad-hoc baselines as well as [20], and highlights the contributions to performance of the different proposed system components (notably, taking the step from fixed to recursive splitting of categories). Performance is indeed best for the proposed method, but considerable margins, and particularly pronounced for tail categories. Presentation ------------ The paper text seems to suffer from a latex compilation issue that results in missing multiple bibliographic and figure references (questions marks instead of numbers: line 79, 176, 239, 277, Tab. 1), which is unfortunate and a possible indication that the paper submission has been rushed or a premature version uploaded. The text, hoever, does contain enough information in order to know what is going on despite not having the references. The authors should please comment on this and what the correct references are in their rebuttal. Apart from this issue, the paper is well written and clearly presents the proposed ideas, experimental results, and relation to prior work. Typo line 74: 'changeling' - challenging; line 96: 'the the'

Reviewer 2



*Approach:* The paper addresses the problem of learning models from a few examples by learning model "dynamics" from the large classes. By simulating the number of training examples in large categories the dynamics are estimated. This is then transferred to categories with few training examples, thereby simulating the presence of many training examples. The approach is implemented as a residual architecture based on the property of compostionality of the transformation as training data is added. The work is related to several recent approaches for meta-learning where a model is trained to predict the parameters of another model based on one or few training examples. This paper presents another way to parameterize the transformation using residual architecture that has some attractive properties. *Results* The approach is compared to predictive baseline [20] on several datasets (SUN, Places, ImageNet). The approach outperforms the sample reweighing baselines as well as [20]. *Evaluation* The paper is well written and presents a number of interesting ideas, which I found very enjoyable. I found the use of the residual architecture to model the diminishing transformations quite interesting. One issue with predictive models (including this work) is that they are limited to small size outputs. This work only considers the last layer weights. How can one generalize the approach to more complex model parameters?

Reviewer 3



This paper presents a work on tackling the challenge of learning classification models from datasets with imbalanced long-tail class distributions. The major idea is to learn a "meta model" to directly estimate the parameters of models trained with many samples using parameters learned on few, or even single, samples. This process is done in a chained manner, which is modeled by residual networks. Classification experiments are performed on image recognition datasets to verify the effectiveness of the proposed approach. Using meta learning to address the problem of imbalanced data distribution is an interesting idea. It clearly differentiates this work from other approaches to solving this problem. However, the authors did not present well this idea. In particular, the writing is not easy to comprehend, and a lot of references are missing in the text (line 176, 239, 240, etc.). In practice, the idea is implemented as residual networks and the meta model is trained to model the dynamics of model parameters when trained on different number of available sample. This is justified by the authors observation on identity mapping and compositionally. Here the authors rely on one key insight that classifier weights for different class share similar dynamics. This is an interesting claim and is crucial for the approach, because it ensures that the dynamics learned on data-abundant classes can be transferred to but we fail to see any analysis or theoretic proof. Another weakness of this work is on the experiment side. The approach is only applied to the last layers of classification networks, which is equivalent to working with linear classifiers. It would be of great interest to see experiments with non-linear models, where this approach could see wide uses. Overall, I believe the authors presents an insightful and interesting idea with potential applications. But in its current status this work is not mature to merit a publication at NIPS. I highly encourage the authors to make significant improvement to the presentation and experiments and resubmit it to another venue.